# Laser Welding of Micro-Wire Stent Electrode as a Minimally Invasive Endovascular Neural Interface

**DOI:** 10.3390/mi16010021

**Published:** 2024-12-26

**Authors:** Bo Wen, Liang Shen, Xiaoyang Kang

**Affiliations:** 1Laboratory for Neural Interface and Brain Computer Interface, Engineering Research Center of AI & Robotics, Ministry of Education, Shanghai Engineering Research Center of AI & Robotics, MOE Frontiers Center for Brain Science, State Key Laboratory of Medical Neurobiology, Institute of AI & Robotics, Academy for Engineering & Technology, Fudan University, Shanghai 200433, China; 22210860062@m.fudan.edu.cn (B.W.); 23210860064@m.fudan.edu.cn (L.S.); 2Ji Hua Laboratory, Foshan 528200, China

**Keywords:** endovascular stent electrode, laser welding, electrochemical impedance spectroscopy (EIS), minimally invasive brain–computer interface

## Abstract

Minimally invasive endovascular stent electrodes are an emerging technology in neural engineering, designed to minimize the damage to neural tissue. However, conventional stent electrodes often rely on resistive welding and are relatively bulky, restricting their use primarily to large animals or thick blood vessels. In this study, the feasibility is explored of fabricating a laser welding stent electrode as small as 300 μm. A high-precision laser welding technique was developed to join micro-wire electrodes without compromising structural integrity or performance. To ensure consistent results, a novel micro-wire welding with platinum pad method was introduced during the welding process. The fabricated electrodes were integrated with stent structures and subjected to detailed electrochemical performance testing to evaluate their potential as neural interface components. The laser-welded endovascular stent electrodes exhibited excellent electrochemical properties, including low impedance and stable charge transfer capabilities. At the same time, in this study, a simulation is conducted of the electrode distribution and arrangement on the stent structure, optimizing the utilization of the available surface area for enhanced functionality. These results demonstrate the potential of the fabricated electrodes for high-performance neural interfacing in endovascular applications. The approach provided a promising solution for advancing endovascular neural engineering technologies, particularly in applications requiring compact electrode designs.

## 1. Introduction

Brain–computer interfaces (BCIs) have gained increasing attention as an innovative technology, showcasing remarkable advances such as controlling exoskeletons via neural signals, operating computer games using brain activity, and applying brainwave recognition for medical diagnoses. Bamdad et al. emphasized that BCIs in neurorehabilitation not only aid motor control but also enhance cognition and psychological well-being. Despite positive experimental and clinical results, challenges such as signal quality, personalized device design, and long-term recording persist [1]. Neuromodulation techniques, especially electrical stimulation, have made notable advancements in treating drug-resistant epilepsy. Fisher et al. suggested that stimulating the anterior thalamus can significantly reduce seizure frequency, providing a new treatment for refractory epilepsy [2]. Similarly, Morrell demonstrated the successful application of responsive cortical stimulation (RCS) in the treatment of partial epilepsy, where RCS monitors brain activity in real-time and provides timely electrical stimulation, effectively reducing the occurrence of seizures [3]. These studies suggest that the broad clinical application of neuromodulation technologies provides substantial support for the further development of BCI systems. Vaid et al. analyzed the application of EEG signals in BCIs, emphasizing that EEG signal processing and feature extraction were key to improving BCI system performance. The time-varying nature and noise interference of EEG signals are major challenges faced by BCIs, requiring improvements in algorithms and techniques to enhance signal accuracy and system stability [4]. Current BCI research focuses on the following three primary types: implantable, wearable, and interventional BCIs. While implantable BCIs offer high-quality neural recordings, their invasive nature and the requirement to open the skull often result in neurologic injuries and low user acceptance. Conversely, wearable BCIs are non-invasive but provide lower spatiotemporal resolution. Ball et al. compared the signal quality of non-invasive and invasive BCI devices, noting that non-invasive devices, although providing lower signal quality, have clear advantages in safety and long-term use, while invasive devices provide higher signal quality but their invasiveness and potential risks limit widespread adoption [5]. Batista et al. explored the role of non-invasive devices combined with vibrotactile feedback in motor imagery training. They demonstrated the unique advantages of non-invasive BCIs in enhancing training effects and improving user experience [6]. Grill et al. discussed the challenges and long-term stability issues faced by implanted neural interfaces, proposing miniaturization and multimodal interface designs to overcome these challenges. While implanted BCIs offer superior signal quality and long-term performance, they still face practical challenges [7]. Cardinale et al. highlighted that while SEEG provides high safety and accuracy for deep brain region localization, its invasive nature introduces certain risks during surgery [8]. Hamer et al. focused on the complications of subdural electrode monitoring, emphasizing the risks of infection and bleeding during implantation, revealing the potential drawbacks of invasive techniques [9]. Tebo et al. noted that despite improvements in cranial epilepsy surgery techniques, invasive procedures still carry risks, particularly during implantation [10]. Käthner et al. investigated the feasibility of combining a P300 BCI with a head-mounted display, demonstrating its potential for rapid communication. This research showed that non-invasive devices could achieve fast and accurate recognition, making them suitable for real-time BCI applications [11]. This further highlights the potential of head-mounted BCIs in interactive communication and real-time applications. Nicolas et al. reviewed the overall development of BCI technology, covering non-invasive, implanted, and interventional devices. The paper thoroughly discussed the advantages and challenges of these devices and predicts that BCI technology will be widely applied in the medical, educational, and entertainment fields in the future [12].

Positioned between these two extremes, interventional BCIs, such as endovascular neural interfaces, combine the advantages of both approaches. These interfaces avoid skull opening, significantly reducing risks while maintaining high signal quality, spatiotemporal resolution, and long-term stability, making them highly promising for scientific and clinical applications. Nowinski et al. simulated and assessed the potential vascular damage caused by deep brain stimulation (DBS) using a 3D atlas, providing a crucial reference for the safety of neural modulation therapies [13]. Thielen et al. emphasized the high accuracy and feasibility of endovascular techniques for neural recording and stimulation, offering new insights for developing more precise and safer brain–machine interface devices [14].

The concept of endovascular stent electrodes originated in 1973 when Penn et al. demonstrated their feasibility through experiments on baboons and provided early evidence of their applicability in humans [15]. Nakase et al. explored the use of guidewires as electrodes for detecting electroencephalogram (EEG) signals within arteries, showing that endovascular electrodes could accurately identify epileptogenic foci and arterial malformations, comparable to subdural electrodes [16]. Stoeter et al. extended this research by placing guidewire electrodes in the middle cerebral artery and frontal–parietal branches of 23 subjects, confirming consistent somatosensory-evoked potentials between endovascular and epidural electrodes [17]. Boniface et al. introduced endovascular EEG technology and its application during the carotid amytal test, demonstrating its effectiveness and reliability in brain function assessment [18]. García et al. focused on the technical aspects of intra-arterial EEG recording, highlighting the importance and potential of this method in neuroscientific research [19]. Kunieda et al. used cavernous sinus EEG to detect the onset and spread of seizures in mesial temporal lobe epilepsy, providing a novel approach for early diagnosis [20]. Mikuni et al. proposed “cavernous sinus EEG”, a new method for the preoperative evaluation of temporal lobe epilepsy, enhancing diagnostic accuracy [21]. Thömke et al. described the use of 16-electrode endovascular EEG recordings during the intracarotid amobarbital test, demonstrating its effectiveness for seizure localization [22]. In 2009, Watanabe et al. introduced highly flexible microwire electrodes designed for endovascular use, reducing size and minimizing immune responses. These electrodes could be delivered into thinner vessels while maintaining efficient signal collection [23]. In 2016, P. Gaba et al. developed a self-expanding electrode array using balloon and basket catheters, achieving tight vessel wall contact [24]. More recently, Fanelli et al. fabricated flexible electrode arrays supported by polycaprolactone (PCL), enabling close vessel wall fitting without a stent [25]. Similarly, Zhang et al. used flexible materials to create electrodes deployable in thin vessels, emphasizing the importance of reducing the size and immune responses for small-animal applications [26]. Neudorfer et al. investigated endovascular deep brain stimulation (DBS) and explored the relationship between vascular structures and DBS targets, providing new insights into neurostimulation therapies [27].

The research by Opie and Oxley et al. further demonstrated the potential of stent-supported endovascular electrodes for chronic experiments in large animals. Their studies showed successful deployment in sheep SSS for long-term neural monitoring, highlighting applications in chronic disease management. They presented chronic EIS of an endovascular stent-electrode array, demonstrating its feasibility and stability for long-term monitoring [28]. They also successfully performed focal stimulation of the sheep motor cortex with a chronically implanted endovascular stent–electrode array, validating the technique’s potential for chronic implantation [29]. At the same time, they discussed the feasibility of chronic, minimally invasive endovascular neural interfaces, laying the groundwork for clinical applications [30]. Also, they used micro-CT and histological evaluation to assess the stability of a neural interface implanted within a blood vessel, confirming its structural and functional viability [31]. Most important, they demonstrated the application of endovascular stent–electrode arrays for high-fidelity, chronic cortical neural activity recordings, proving its advantages in long-term neural recording [28,32]. However, electrode diameters of 400–750 μm limited their use to large vessels, restricting compatibility with smaller animals.

In this study, these advances are built on by investigating the feasibility of fabricating a micro-wire stent electrode as small as 25 μm through laser welding. Using this technique, the metal wire was securely joined to a 300 μm platinum sheet without compromising its structural integrity, significantly minimizing the heat affect zone and material modifications typically caused by resistive welding. The resulting device could be retracted into a catheter with an inner diameter of 0.6 mm, enabling deployment in small animals and thin vessels. The potential of laser ablation technology to produce minimally invasive, high-performance neural interfaces for applications in both scientific research and clinical settings is demonstrated in this work.

## 2. Materials and Methods

### 2.1. Simulation

#### 2.1.1. Distribution of the Working Electrode Simulation

The positioning of electrodes plays a crucial role in both electrical stimulation and signal recording, particularly the arrangement of the working and ground electrodes. In this study, we used COMSOL Multiphysics 6.1 to simulate the optimal relative positioning of stent electrodes. Building on prior research, we evaluated the efficacy of electrode placement for endovascular stimulation by applying electrical currents to the electrode sites and calculating the resulting potentials at the vessel boundaries [33].

In the simulation, to better guide the actual electrode preparation process, we set the simulation materials to have the same physical properties as the real materials. Nitinol (NiTi) alloy with 55% nickel and 45% titanium was selected as the stent material, platinum foil with a purity of 99.99% as the electrode material, blood as the internal medium of the blood vessel, and cerebrospinal fluid as the external medium. The boundary conditions were defined as follows: the grounding electrode was a rectangular pad measuring 7 mm × 2 mm × 30 mm. The working electrodes were assigned a fixed potential of 1 V, while all other boundaries were set to electrically insulated conditions.

Figure 1 illustrates the simulation results for the relative positioning of micro-wire stent electrodes. The analysis included a comparison of 3D surface streamline potential maps and potential distribution maps for different grounding electrode configurations. At a working electrode plane with a 10 mm radius, the maximum potential values were evaluated for each model. The uniform distribution electrode (Model A) showed a maximum potential of 0.72 V, comparable to the designed distribution electrode (Model B) at 0.78 V. These findings suggest that Model A performs similarly to Model B. However, Model B offers the advantage of recording more vascular electrocorticography signals simultaneously. In contrast, using the electrode on the stent as the grounding electrode reduces the available channels for the working electrodes, making the rectangular pad the preferred grounding configuration.

The simulations of the electrical stimulation effectively demonstrated the spatial relationship between the electrodes and the stent, as well as the design principles for the electrode system. Previous studies have explored the distribution of electrodes in blood vessels [33] and analyzed electrical stimulation in various animal models, such as mice [34]. While these studies suggested a practical 3-electrode arrangement around the stent, our findings indicate that a 4-electrode configuration can record neural signals across a wider range. Additionally, in our study, we incorporated electrodes positioned at varying horizontal planes to better reflect real-world conditions.

#### 2.1.2. Laser Welding Simulation

In this study, we investigated the significance of simulating the heat-affected zone in laser welding, highlighting that this simulation tool not only optimized process parameters but also accurately predicted potential changes in material properties. The application of simulation techniques significantly improved welding quality and consistency, particularly when addressing complex geometries or extreme environmental conditions. Moreover, this method effectively reduced experimental costs, providing a reliable reference for research. Additionally, in this study, the remarkable potential of this approach in the development of new materials and welding technologies was demonstrated, offering a scientific foundation for process optimization and technological advancements.

In this study, the laser heat source was primarily modeled using a two-dimensional Gaussian distribution, with the main calculation formula as follows:$$q\left(r,z\right)=\frac{2P}{\pi r_{0}^{2}}\exp\left(-\frac{2r^{2}}{r_{0}^{2}}\right)\exp\left(-\frac{\alpha z}{k}\right)$$
where q(r,z) represents the power density of the heat source at a specific point in space ($W/m^{2}$); P is the total laser power; r is the radial distance from the laser’s focal point (m); $r_{0}$ represents the laser beam radius; z is the position in the depth direction; α is the absorption coefficient; and k is the thermal conductivity.

Combining the material heat conduction equation to obtain the distribution of temperature field, as follows:$$\rho c\frac{\partial T}{\partial t}=k\nabla^{2}T+q(r,z)$$

For the base material, we define two heat fluxes, the convective heat flux and generalized inward heat flux. The convective heat flux describes the heat exchange caused by the temperature difference between the workpiece surface and the environment. This heat exchange is primarily driven by the surrounding medium carrying heat away through the convection process, and its expression is as follows:$$q_{conv}=h(T_{s}-T_{\infty })$$

In laser welding, the surface temperature can be extremely high (much higher than the ambient temperature), and the convective heat flux significantly impacts the cooling rate at the boundaries of the heat-affected zone, thereby influencing the microstructure and mechanical properties of the material. The generalized inward heat flux is typically used to describe the energy density input by the laser beam as a heat source into the workpiece, accounting for laser energy absorption, scattering, and the thermal properties of the material. Its typical expression is as follows:$$q_{\mathrm{in}\;}=\alpha I_{0}exp(-\frac{2r^{2}}{r_{0}^{2}})exp(-\beta z)$$

This heat flux characterizes the energy distribution of the laser beam as it penetrates from the surface into the material, typically following a Gaussian distribution. It is the core of heat conduction simulation in laser welding, directly determining the shape, depth, and extent of the heat-affected zone and the molten pool. The convective heat flux is the primary mechanism for surface heat dissipation, whereas the generalized inward heat flux serves as the source of heat conduction within the material. Together, they influence the dynamic equilibrium of the temperature distribution; when the heat input from the laser exceeds the surface heat dissipation, the molten pool expands, and the heat-affected zone increases. Conversely, if the convective heat dissipation is too strong, the molten pool shrinks, and the material cools rapidly.

The coupling of these two fluxes is typically achieved in simulation models by solving the following heat conduction equation:$$\rho c\frac{\partial T}{\partial t}=k\nabla^{2}T+q_{in}-q_{conv}$$

The simulation results are as shown in Figure 2.

Based on the simulation results, we designed and validated an optimal arrangement for the electrode array, ensuring maximum performance during the laser welding process while considering the thermal effects on material properties. Furthermore, by simulating the interaction between the laser and the material during welding, we conducted an in-depth analysis of heat transfer, melt pool formation, and material modification induced by the laser. These simulation studies provide theoretical guidance for the implementation of specific experiments, enabling us to optimize critical parameters such as laser power, current intensity, and electrode arrangement in practical operations to achieve the best balance between performance and process efficiency.

### 2.2. Orthogonal Experiment

In this research, an orthogonal experimental design was employed to investigate multiple factors and levels by selecting representative combinations based on the principle of orthogonality. A multi-level orthogonal table was constructed for three laser welding parameters to identify the most effective cutting conditions, as presented in Table 1. In order to better weld polyimide-coated platinum-tungsten micro-wires (92%/8%) with the platinum disk with a purity of 99.99%, we first performed laser ablation on the metal wires in this study. The surface insulation layer was ablated by laser cutting, and the femtosecond laser processing parameters were optimized through orthogonal experiments. The specific orthogonal experiment table is shown in Table 2. In this study, cutting performance was evaluated using both microscopic imaging and electrochemical impedance spectroscopy (EIS), enabling the determination of the optimal parameter combination under varying conditions.

Microscopic images were analyzed to assess the precision of the cutting lines and the effectiveness of wire insulation layer removal. These images revealed the surface morphology post-laser cutting, including the clarity of the cut lines and any material damage. Additionally, EIS measurements provided essential insights into the electrochemical behavior of the electrodes. By examining impedance values across different frequencies, the electrodes’ performance in electrochemical reactions was evaluated in this study. The EIS results reflected the condition of the electrode surface, interface properties, and conductivity, serving as critical indicators for identifying the best combination of laser welding parameters.

### 2.3. Laser Welding

In this study, a 1030 nm femtosecond laser was employed to precisely ablate insulated nanowires for improved welding. This laser was chosen for its high precision and minimal thermal impact during the ablation process. After parameter optimization, the femtosecond laser effectively removed the insulation layer without causing damage to the nanowires. By exposing 2 mm of polyimide-coated platinum-tungsten micro-wires (92%/8%), the process ensured that the electrodes remained uncontaminated by the insulation layer during laser welding, thereby maintaining their functionality.

Traditional stent electrodes were created using resistive welding between a wire and a platinum disk, which restricted their use primarily to large animals or thick blood vessels. The platinum disk typically required a diameter of 400–750 μm to accommodate the resistive welding process [29,32]. However, this design made it nearly impossible for the stent electrode to be retracted into a catheter with an inner diameter as small as 0.6 mm. The design in our study was the diameter of 300 μm platinum disk. To address this limitation, a laser welding technique was developed to weld the platinum disk with the micro-wire. This innovation significantly reduced the electrode diameter to just 300 μm, enabling its retraction into thinner catheters.

### 2.4. Fabrication

The main assembly process for the laser-welded stent electrode is depicted in Figure 3. Initially, a laser cutting technique was utilized to precisely ablate the insulated micro-wires. The electrodes were fabricated from 25 μm polyimide-coated platinum-tungsten (92%/8%) wires (Goodfellow, Huntingdon, UK) using laser processing with the diameter of 300 μm and the thickness of 50 μm platinum disk. These micro-wires were then inserted into a catheter with an inner diameter of 0.6 mm and an outer diameter of 0.8 mm, allowing them to extend out from the opposite end of the catheter. To prevent short circuits or direct contact between the electrodes and the stent, a 2 μm-thick parylene-C film was deposited onto the stent. For this purpose, a commercially available Solitaire stent (Solitaire SAB; Covidien, CA, USA) with an expanded outer diameter of 3 mm was used.

Subsequently, the electrodes were bonded to the stent using a UV-curable adhesive (Dymax 1128-A-M, UV Pacific, Edwardstown SA, Australia), which was cured under 365 nm UV light for 20 s. During the bonding process, care was taken to ensure uniform application of the adhesive to the contact area between the electrode and the stent. Each electrode was positioned precisely according to the results obtained from simulations. The fabricated laser welding endovascular stent electrodes are shown in Figure 4.

At last, the stent electrode was carefully retracted into the catheter. This procedure avoided the steps of dismantling the stent and bonding to a stainless steel guide wire, thereby simplifying the assembly process. This method was also helpful in reducing the potential damage of the stent electrode during the implantation experiment.

### 2.5. Calculation of Charge Storage Capacity and Charge Injection Capacity

The electrochemical characterization of the laser welding electrode was performed in saline, using an Ag/AgCl electrode as the reference electrode and a large-area tungsten wire as the grounding electrode [35]. Electrochemical impedance spectroscopy (EIS) was measured by applying a 10 mV sine wave across a frequency range of 1 Hz to 100 kHz. The charge storage capacity (CSC) was derived from cyclic voltammetry (CV) curves, with testing conditions set between −0.6 and 0.8 V and a scan rate of 0.1 V/s. CSC was calculated using the following formula:$$CSC=\frac{1}{v}\int_{E_{c}}^{E_{a}}\left|\frac{i}{S}\right|dE$$
where E is the electrode potential; i is the detected current; S is the geometric surface area of the working electrode; v is the scan rate; and $E_{c}$ and $E_{a}$ are the anodic and cathodic voltage limits [36,37].

The charge injection capacity (CIC) was determined through voltage transient analysis. A current pulse was applied between the laser-welded electrode and the ground electrode, while an oscilloscope recorded the voltage response between the laser-welded electrode and the reference electrode. To determine the maximum polarization voltage, the amplitude of the bidirectional symmetric current pulse was adjusted. The current pulse had a width of 300 μs, an interval of 30 μs, and a frequency of 50 Hz. The maximum polarization position was identified when the negative polarization reached the water reduction potential (−0.6 V) or the positive polarization reached the water oxidation potential (+0.8 V). At this point, the injected charge was calculated as the product of current amplitude and pulse width, then normalized by the electrode’s geometric surface area to determine the charge injection capacity. The calculation formula is as follows:$$CIC=\frac{Q}{A}=\frac{I\times t}{S}$$
where I is the amplitude of the current pulse; t is the width of the current pulse; and S is the geometric surface area of the electrode.

## 3. Results and Discussion

### 3.1. Simulation

In this study, we investigated the relative positioning of the laser-welded stent electrodes, focusing on comparisons of 3D streamline potential maps and potential distribution maps under varying grounding electrode configurations. For a working electrode plane with a 10 mm radius, the maximum potential values were observed to be 0.72 V for the uniform distribution electrode (Model A) and 0.78 V for the designed distribution electrode (Model B). These results indicate that both configurations deliver a comparable performance. However, Model B demonstrated a distinct advantage by enabling the recording of a greater number of vascular electrocorticography signals. Conversely, using the stent electrode as the grounding electrode reduced the available channels for working electrodes. Thus, the rectangular pad configuration was the optimal choice for grounding, balancing functionality and signal acquisition capacity.

The laser welding simulation results demonstrated the effectiveness of the applied model in predicting the heat-affected zone (HAZ) and melt pool dimensions. The three-dimensional simulation captured the spatial distribution of the HAZ and melt pool under varying laser cycles, the result provided cross-sectional views. With an increasing number of laser cycles, the melt pool size on the platinum plate progressively expanded, reflecting the accumulation of thermal energy and its diffusion into the material. The findings confirmed that when the laser heat input exceeded surface dissipation, the melt pool expanded, increasing the HAZ. Conversely, strong convective dissipation led to rapid cooling, resulting in a reduced melt pool. So, when conducting the process, it is advisable to minimize the frequency of the laser or increase the time span between laser welding pulses as much as possible. Therefore, in this study, the frequency of the laser welding was mainly used at 1 Hz.

Building on the simulation results, an optimal electrode arrangement was designed and validated. Also, we used the simulation to instruct our orthogonal experiment.

### 3.2. Laser Welding and Laser Ablation

Based on the orthogonal experiment and the results of laser cutting, we refined the range of processing parameters. We employed a laser energy pulse of 19 μJ and a power of 18 W, and the divider was 25. Based on this, we compared the reduced laser energy, processing repetition, and marking speed. In Figure 5, we observe that at 5% laser energy, with eight processing cycles and a marking speed of 100, the electrochemical impedance value at 1 kHz is minimized. Therefore, we conducted further experiments to refine the number of processing cycles based on this result. This analysis confirmed that laser processing parameters could effectively optimize impedance characteristics. It proved that laser processing can modify metal insulation wires to enhance their electrochemical performance.

In the end, the laser cutting parameters were energy 19 μJ, scaling factor 5%, and the frequency divider 25. The marking speed and jump speed were both set to 200× 200. Figure 6 shows the laser cutting image. Figure 7 shows the laser welding effect of the platinum disk and the micro-wire.

In this study, we compared the effects of different laser welding parameters on the electrochemical performance of electrodes. The results indicate that the optimal laser welding parameters require a relatively fixed range for pulse width and spot size. During the actual welding process, an excessively long laser pulse width may result in the melting of the platinum sheet, which is a phenomenon we aimed to avoid based on our simulation experiments. Moreover, when the spot size is too small, the welding between the insulating wire and the platinum sheet is not effectively achieved. Therefore, under the conditions of a pulse width of 0.2 ms and a spot size of 0.2 mm, in this study, a detailed examination was conducted of current intensity parameters to determine the optimal combination.

Based on the orthogonal experiments, we found that the most optimal pulse width and spot size for laser welding were 0.2 ms and 0.1 mm, which minimized the heat-affect zone and met the requirements for welding micro-wire to platinum disk. The EIS data of laser welding are shown in Figure 8. The averaged CV curve and voltage transient curve of the electrode are shown in Figure 9.

The results indicate that as the current gradually increases from 12 A, the impedance of the electrode initially decreases and then increases. This trend demonstrates the significant influence of current intensity on the quality of laser welding. By analyzing the impedance values under different current intensities, we identified that the optimal welding condition occurs at a current of 17 A, where the electrode impedance reaches its minimum. Therefore, adopting a current intensity of 17 A in the experiments can effectively optimize the welding performance and enhance the electrochemical characteristics of the electrode.

As the results in Table 3 demonstrate, the electrodes processed through laser ablation and laser welding exhibit outstanding electrochemical performance, comparable to that of traditional stentrodes [29] with significantly larger surface areas. By optimizing laser processing parameters such as energy, repetition cycles, and marking speed, we achieved the precise fabrication of miniaturized electrodes, which not only significantly reduced their size but also maintained high electrochemical efficiency. The electrode diameter prepared in this study is only 300 μm, greatly improving the ues of the internal space of the catheter.

As shown in Table 4, the maximum tensile force required to break the electrode before and after laser welding does not differ significantly, being 0.93 N, which is sufficient to prove that the electrodes fabricated by laser welding exhibit good mechanical properties.

## 4. Conclusions

In this study, a novel approach is demonstrated for fabricating endovascular micro-wire stent electrodes using laser welding and ablation technologies. The method significantly reduces the electrode size, making it suitable for narrower blood vessels. The quantitative results highlight the excellent electrochemical performance of the electrodes fabricated under optimal laser welding parameters, with a 1 kHz impedance of 4117 Ω, a 1 kHz phase of −68.23 degrees, a charge storage capacity (CSC) of 8.745 ($mC/{cm}^{2}$), and a charge injection capacity (CIC) of 1.617 $\times 10^{-4}C/{cm}^{2}$. These results indicate that the electrodes possess excellent stability and suitability for use as neural interfaces in confined vascular environments.

The integration of laser welding not only ensures the mechanical strength of the electrodes, with a maximum tensile strength of 0.93 N, but also optimizes their electrochemical properties. These findings underline the critical role of laser welding in enhancing both mechanical and electrochemical performance. Furthermore, in this study, optimal laser processing parameters are identified, providing a strong foundation for future developments in laser-based fabrication methods. This approach offers a promising pathway for advancing intravascular stent electrodes and expanding their applications in neural engineering.

In the future, with the continued advances in laser technologies and the increasing demand for high-precision processing, the laser welding-based microscale electrode fabrication process is expected to find applications in broader biomedical fields. For instance, by optimizing processing parameters and material selection, the biocompatibility and long-term stability of the electrodes can be further enhanced. Incorporating a combined thermomechanical analysis to improve the modeling and understanding of laser ablation physics is also important. Moreover, integrating this technology with intelligent monitoring systems holds the potential to develop multifunctional minimally invasive devices capable of real-time recording and stimulation of neural signals, thereby promoting the application of neural interface technologies in disease diagnosis, treatment, and neural repair.

## Figures and Tables

**Figure 1 micromachines-16-00021-f001:**
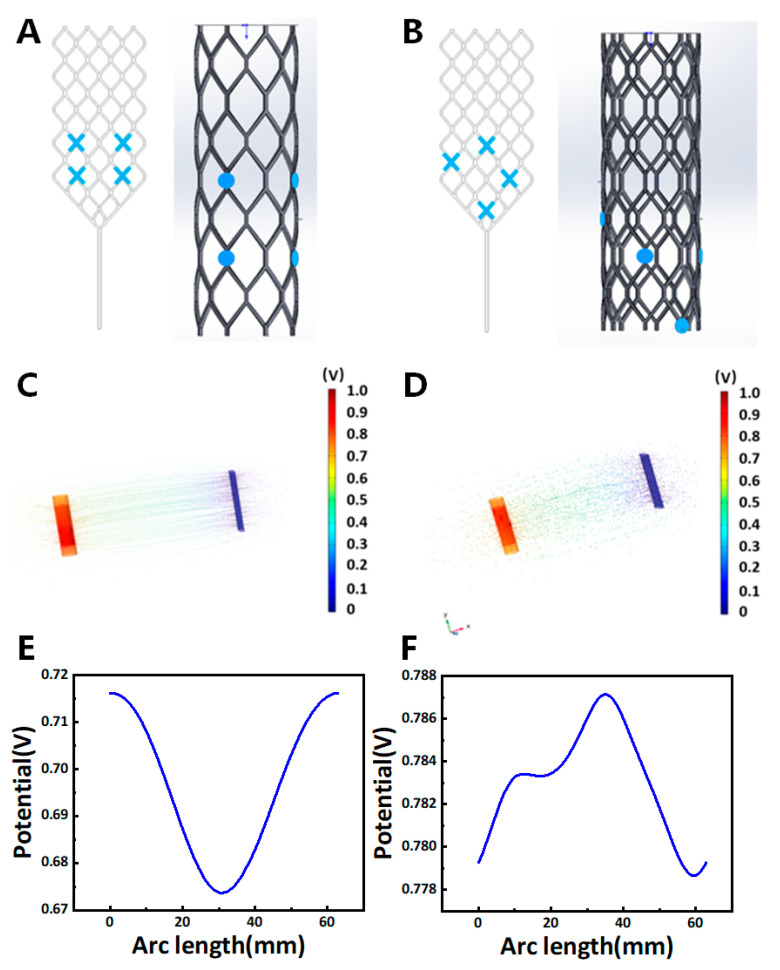
The simulation of the relative position design for the micro-wire stent electrode employed three different models to evaluate electrode configurations. (**A**,**B**): the blue “×” and disks represent the positions of the working electrodes. In models A and B, the grounding electrode is configured as a large rectangular pad measuring 7 mm × 2 mm × 30 mm; (**C**,**D**): the 3D surface streamline potential maps display the potential distribution after electrical stimulation in each of the three models; (**E**,**F**): the potential distribution maps correspond to the working electrode plane, which is defined as a circular area with a radius of 10 mm.

**Figure 2 micromachines-16-00021-f002:**
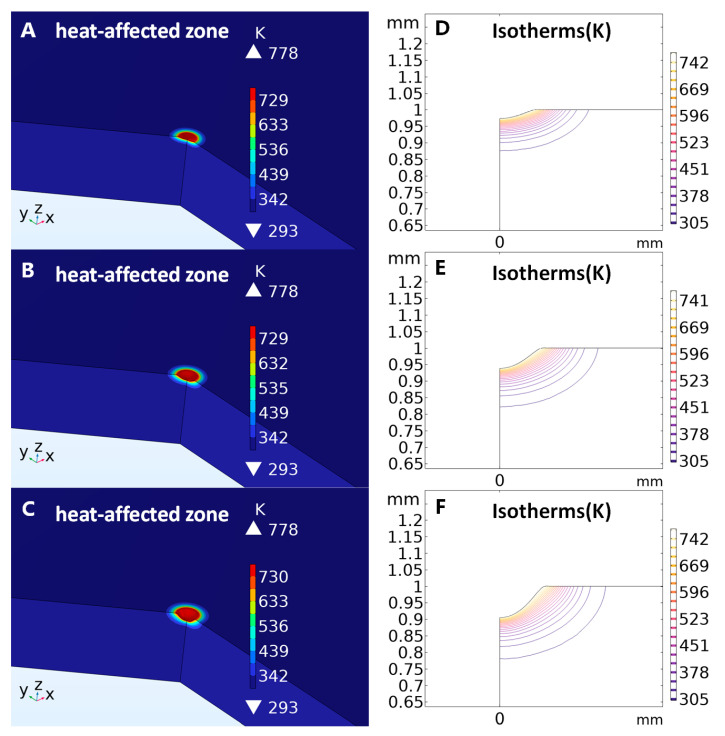
(**A**–**C**) illustrate the three-dimensional heat-affected zone and melt pool dimensions during laser welding on a platinum disk, whereas (**D**–**F**) present two-dimensional cross-sectional views of the melt pool. It is observed that with an increasing number of laser cycles, the melt pool size on the platinum disk progressively expands.

**Figure 3 micromachines-16-00021-f003:**
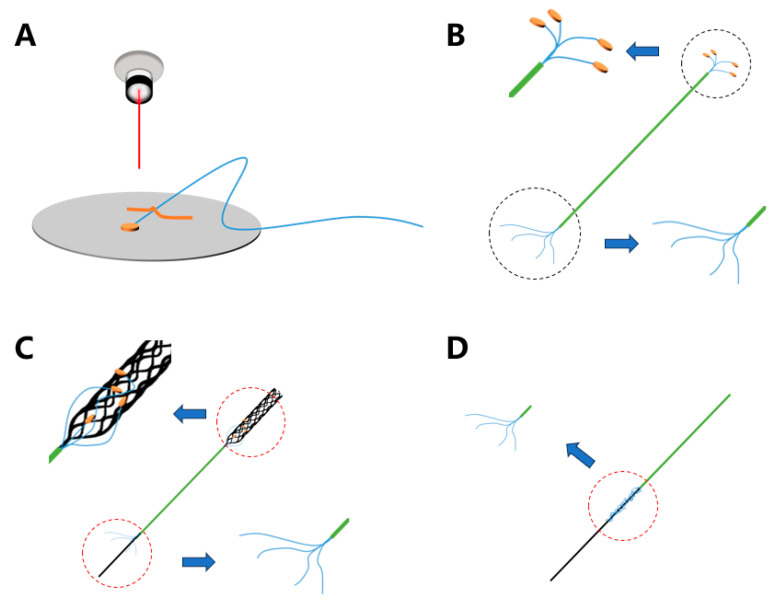
The main steps (**A**–**D**) of the fabrication of the laser welding endovascular stent electrode.

**Figure 4 micromachines-16-00021-f004:**
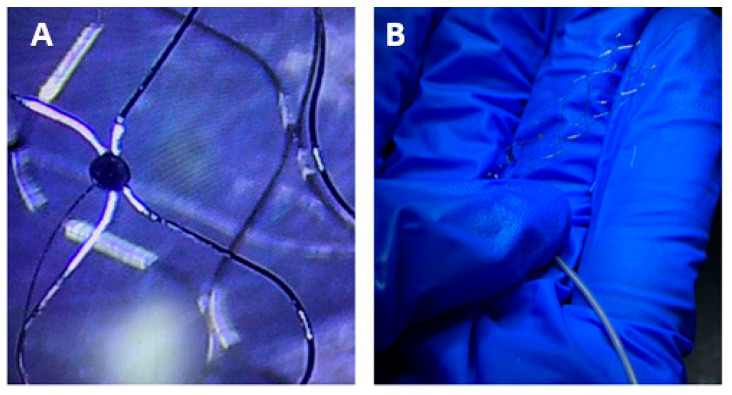
The images of the fabricated laser welding endovascular stent electrode. (**A**) present the microscopic image of the bonding effect of the laser welding electrode with the stent. (**B**) present the normal image of the bonding effect.

**Figure 5 micromachines-16-00021-f005:**
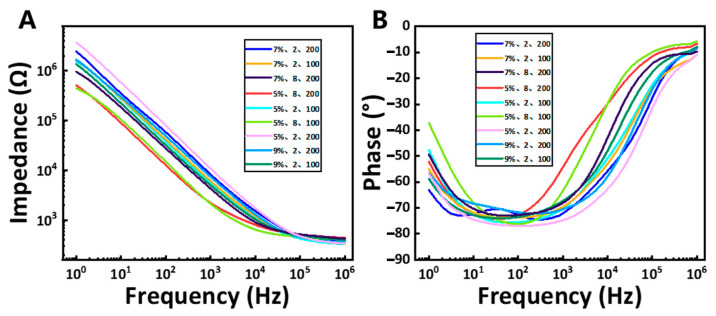
(**A**,**B**) Averaged EIS at different processing parameters (n = 6). It indicates the optimal choice for laser ablating the micro-wire. The impedance decreases correspondingly with the decrease in laser energy and the increase in processing times. At the same time, increasing the laser cutting marking speed was equivalent to increasing the laser energy.

**Figure 6 micromachines-16-00021-f006:**
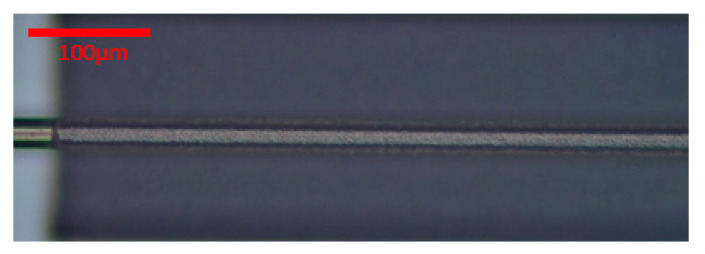
The laser ablation image of the micro-wire under the optimal parameter.

**Figure 7 micromachines-16-00021-f007:**
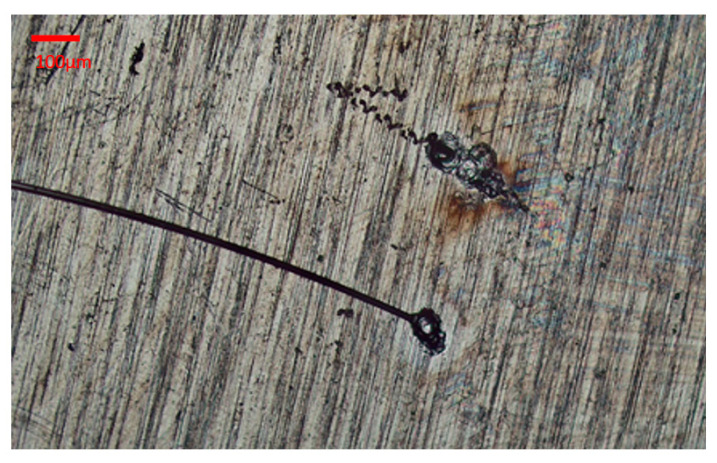
The laser welding image of the micro-wire with the platinum disk under the optimal parameter.

**Figure 8 micromachines-16-00021-f008:**
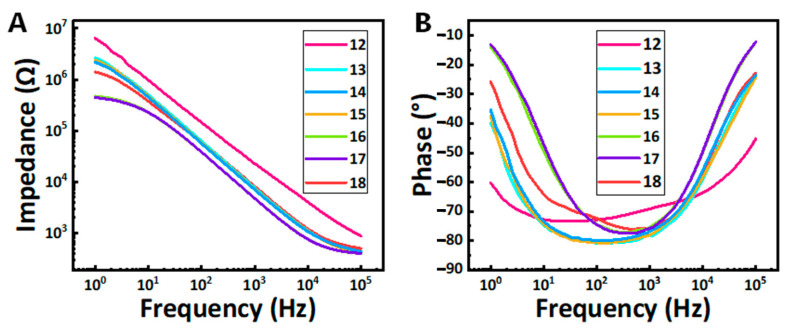
Averaged EIS of the laser welding endovascular stent electrodes at different current intensity (n = 6) (**A**) present the impedance of the electrode after the laser welding. (**B**) present the phase angle of the electrode after laser welding.

**Figure 9 micromachines-16-00021-f009:**
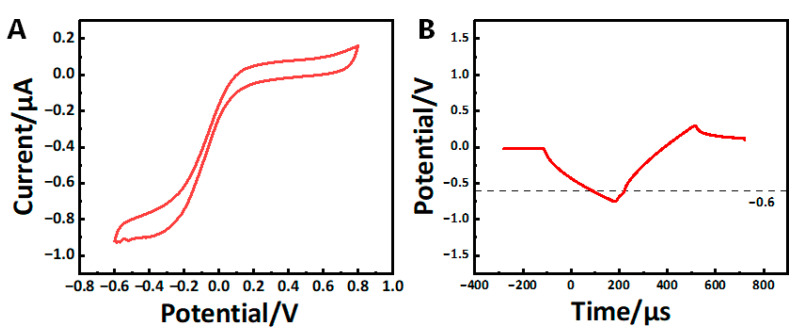
(**A**) Averaged CV curve of the electrode. (**B**) Averaged voltage transient curve of electrodes (n = 6).

**Table 1 micromachines-16-00021-t001:** The multi-level orthogonal experimental table for the three laser welding factors.

Experimental Label	Current Intensity (A)	Pulse Width (ms)	Spot Size (mm)	Impedance at 1 kHz (Ω)
1	10	0.1	0.1	Not welded
2	10	0.2	0.2	8323
3	10	0.3	0.3	5229
4	15	0.1	0.1	6796
5	15	0.2	0.2	4204
6	15	0.3	0.3	Melt through
7	20	0.1	0.1	7254
8	20	0.2	0.2	5170
9	20	0.3	0.3	Melt through

**Table 2 micromachines-16-00021-t002:** The multi-level orthogonal experimental table for the three laser cutting factors.

Experimental Label	Laser Energy	Repetition	Marking Speed	Impedance at 1 kHz (Ω)
1	5%	2	100	4710
2	5%	8	200	1997
3	5%	16	100	Broken
4	7%	2	200	7134
5	7%	8	100	Broken
6	7%	16	200	Broken
7	9%	2	100	4360
8	9%	8	200	Broken
9	9%	16	100	Broken
10	5%	2	200	9317
11	5%	8	100	1875
12	5%	16	200	Broken
13	7%	2	100	5330
14	7%	8	200	3734
15	7%	16	100	Broken
16	9%	2	200	6460
17	9%	8	100	Broken
18	9%	16	200	Broken

**Table 3 micromachines-16-00021-t003:** The electrochemical data of laser welding electrodes under optimal parameters.

1 kHz Impedance (Ohm)	1 kHz Phase (−Degree)	$\mathrm{CSC}\;(mC/{cm}^{2})$	$\mathrm{CIC}\;(\times 10^{-4}C/{cm}^{2})$
4117	68.23	8.745	1.617

**Table 4 micromachines-16-00021-t004:** The maximum tensile force required to break the laser welding electrodes (n = 6).

Before Laser Welding	After Laser Welding
1.32 N	0.93 N

## Data Availability

The data presented in this study are available on request from the corresponding author.

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
