# Peer review of "Laser Welding of Micro-Wire Stent Electrode as a Minimally Invasive Endovascular Neural Interface"

_micromachines, 2024, doi:10.3390/mi16010021_

Round 1

Reviewer 1 Report

Comments and Suggestions for Authors

The manuscript is devoted to the study of Laser Welding of Micro-Wire Stent Electrode. The study is relevant and in demand in practice. When reviewing the manuscript, I had several recommendations and comments for the authors.

1. The introduction is very short, does not contain a detailed analysis of previously performed studies. The authors of the manuscript often use group references, for example 1-4, 5-9, 10-14, 27-32. I recommend that the authors analyze in more detail each publication they refer to.

2. In section 2. with the title Materials and Methods, very little attention is paid to the materials. It is indicated that nitinol and platinum are used. What is the chemical composition of these materials? Is it pure platinum? Specify the chemical composition of Nitinol.

3. The figures in the manuscript have a significantly different design. In Figures 1 and 2, the numbers are very small. It is necessary to increase the font size and provide inscriptions on the scales. What units of measurement are on the scale in Fig. 1 (C, D)?

4. In Fig. 2 (A, B, C) the numbers are shown in black. They are impossible to read. The welding zone can be enlarged by means of a callout. The font size of the numbers is very small.

5. I recommend rewriting the explanatory captions of the Figures. "It is observed that, with an increasing number of laser cycles, the melt pool size on the platinum plate progressively expands." In my opinion, such expressions should not be in the captions.

6. The Conclusions say "the fabricated endovascular microwire stent electrodes demonstrated stable mechanical strength and excellent electrochemical performance both in vitro." and "The results highlighted the critical role of laser welding in optimizing the mechanical and electrochemical properties of stent electrodes". The manuscript did not provide the results of the evaluation of the mechanical properties. Please clarify this situation.

7. The Conclusions contain general phrases and do not contain quantitative results. The first and second paragraphs of the Conclusions are essentially the same. In my opinion, the conclusions need to be reformulated.

Reviewer 2 Report

Comments and Suggestions for Authors

Well written paper, here are some comments to further improve the quality of the paper: 

1. Can you explain more why other heat dissipative mechanisms like radiative heat transfer don't play an important role in the thermal analysis of the laser welding process ?

2. Do the temperatures get high enough to cause some vaporization of the material ?  this has not been discussed in the paper.

3. Laser ablation is a combination of a thermal process due to the intense heat generated in the heated zone as well as a mechanical process due to the thermomechanical shock of the ablation. Such systems are better modeled using a thermomechanical analysis considering thermoelastic effects as well which can modify the mechanical properties of the material after ablation. The analysis in the paper does not talk about the thermomechanical effects, adding this analysis can give a more accurate representation of the ablation physics involved.

4. It would be helpful to have an extra figure depicting a cross section of an individual micro wire showing the insulation (before laser welding) and what the cross section would look like post welding. 

Round 2

Reviewer 1 Report

Comments and Suggestions for Authors

The authors of the manuscript responded to my comments in detail. They made edits to the manuscript. I recommend this article for publication in this version.